# A New Synthetic Aperture Radar (SAR) Imaging Method Combining Match Filter Imaging and Image Edge Enhancement

**DOI:** 10.3390/s18124133

**Published:** 2018-11-26

**Authors:** Bing Sun, Chuying Fang, Hailun Xu, Anqi Gao

**Affiliations:** School of Electronics and Information Engineering, Beihang University, Beijing 102200, China; fussy@buaa.edu.cn (C.F.); xuhailun@buaa.edu.cn (H.X.); Gaoaq@buaa.edu.cn (A.G.)

**Keywords:** imaging, edge detection, image segmentation, back-projection algorithm, differential operation

## Abstract

In general, synthetic aperture radar (SAR) imaging and image processing are two sequential steps in SAR image processing. Due to the large size of SAR images, most image processing algorithms require image segmentation before processing. However, the existence of speckle noise in SAR images, as well as poor contrast and the uneven distribution of gray values in the same target, make SAR images difficult to segment. In order to facilitate the subsequent processing of SAR images, this paper proposes a new method that combines the back-projection algorithm (BPA) and a first-order gradient operator to enhance the edges of SAR images to overcome image segmentation problems. For complex-valued signals, the gradient operator was applied directly to the imaging process. The experimental results of simulated images and real images validate our proposed method. For the simulated scene, the supervised image segmentation evaluation indexes of our method have more than 1.18%, 11.2% and 11.72% improvement on probabilistic Rand index (PRI), variability index (VI), and global consistency error (GCE). The proposed imaging method will make SAR image segmentation and related applications easier.

## 1. Introduction

In the last several decades, synthetic aperture radar (SAR) has attracted much attention for its application to radar imaging due to the need for the all-weather observation of Earth. SAR imaging algorithms and image processing are two significant research topics. The studies on SAR imaging algorithms have examined many algorithms, such as the back-projection algorithm (BPA), chirp scaling algorithm (CSA), range-Doppler algorithm (RDA), and so on [1,2]. The BPA is commonly applied to obtain the most accurate result under arbitrary trajectories. In fact, for the formation of some special images, the BPA is the only option. However, the BPA procedure is quite time-consuming, and there have been many attempts to improve the speed of this algorithm [3,4,5]. On the other hand, other algorithms use various approximations to accelerate the speed. Namely, the range-Doppler, Chirp Scaling, and other imaging algorithms, all of which are faster than the BPA and can also achieve accurate results. Aside from image formation, there are various other potential SAR image applications, including edge detection, speckle reduction, image segmentation, and target recognition. Among them, SAR edge detection and image segmentation are the two fundamental procedures. Different from most of the popular methods, in the method proposed herein, SAR imaging is fused with a first-order gradient operation and image segmentation to improve SAR detection. In general, SAR image edge detection methods can be divided into a variety of filtering approaches, hypothesis testing methods, and other similar procedures. The commonly used edge detection operators include the Roberts, Sobel, Prewitt, Laplacian, and Canny operators, among many others [6]. Meanwhile, for SAR image segmentation, the most frequently utilized methods are the Markov random fields method [7], partial differential equation method [8], and clustering method [9]. However, SAR edge detection and image segmentation can still only achieve satisfactory results using the original SAR images, and with difficulty, due to SAR image characteristics. To overcome this issue, some algorithms have been proposed to improve the SAR image results. For instance, in [10,11,12], the authors suggested some algorithms for SAR image edge detection. Moreover, in [13,14,15,16], the authors provided algorithms for SAR image segmentation. However, these algorithms were built upon characteristic SAR images and worked under specific conditions and assumptions. In addition, to improve SAR imaging, Yanik and Li proposed an approach for the direct segmentation of SAR images, which was a kind of filter back-projection (FBP) [17]. On the other hand, Pena, Garza, and Qiao [18,19] successfully processed real echo data using the algorithm.

In this paper, we propose a new imaging method that combines SAR image reconstruction and image processing. SAR echo data usually contain a lot of noise, so it is difficult to proceed with raw echo data directly. However, the combination of SAR imaging and image processing could mitigate this problem during SAR application. In this study, we used the BPA approach because of its high precision [20]. Furthermore, by using the mentioned SAR image edge detection and image segmentation methods on the obtained edge-enhanced SAR image, the processed result can be improved further. Although the obtained image contains complex-valued data, the edge detection and image segmentation algorithms were implemented using real-valued data, i.e., the amplitude of the complex-valued SAR image.

This paper is organized as follows. First, we provide a brief introduction to the proposed algorithm. Second, we explain the background of the traditional BPA, the gradient operation for edge detection, the selected edge detection, and the image segmentation algorithms. Then, we describe the proposed algorithm in detail. Third, we provide the evaluation index and simulation results to demonstrate the effectiveness of our algorithm. Next, we discuss the advantages and disadvantages of the proposed algorithm. The last paragraph concludes the paper.

## 2. The Traditional Algorithm

In this section, the basic concepts and terminology related to SAR image processing are introduced, including the traditional back-projection algorithm (BPA), gradient operators, and selected SAR edge detection and segmentation algorithms.

### 2.1. Back-Projection Algorithm

To better understand the proposed algorithm, we first introduce the traditional BPA. The model of SAR echo data can be expressed by
(1)$$d\left(t,s\right)\approx \int_{z\in target}V(z)A\left[z,x_{0}\left(s\right),\overset{⌢}{e}\;\left(s\right)\right]e^{i\pi \gamma {\left[t-t_{n}\left(s\right)\right]}^{2}}dz$$
where *t* is the fast time, *s* is the slow time, V(z) is the reflectivity function of the target, $A[z,x_{0}\left(s\right),\overset{⌢}{e}\;\left(s\right)]$ is the weight function corresponding to the antenna, $x_{0}\left(s\right)$ is the antenna position, $\overset{⌢}{e}\;\left(s\right)$ is the normalized vector of the antenna, γ is the chirp ratio, and $t_{n}\left(s\right)$ is the time delay of the *n*-th target point. The issue of image reconstruction lies in solving V(z) from the echo data. Based on imaging theory, the model of a SAR image can be expressed by
(2)$$I\left(z_{t}\right)=\;\int \int \;e^{iw2|z_{t}-x_{0}\left(s\right)|/c}Q\left(w,s,y\right)D\left(w,s\right)dwds$$
where $z_{t}$ is the point target position in the fast time domain, *w* is the frequency from the spectrum, *y* is the coordinate, and D(w,s) is the echo data expression (d(t,s)) in the frequency domain. The reflectivity function V(z) can be solved by Equation (Equation 2).

D(w,s) can be represented as follows:(3)D(w,s)=∫∫A(w,s,z)V(z)dz
where A(w,s,z) includes γ. Considering the echo data, we can achieve the following [21]:(4)$$I\left(z_{t}\right)=\;\int \int \;e^{2ik(|R_{s,z_{t}}|-|R_{s,z}|)}Q\left(w,s,z\right)A\left(w,s,z\right)V\left(z\right)dwdsdz$$
where $R_{s,z}=z-x_{0}$ refers to the distance between the antenna and the target, while $R_{s,z_{t}}=z_{t}-x_{0}$ represents the distance between the antenna and a specific point in the imaging area. According to [21], one of the key properties in the imaging method is to form
(5)$$\begin{array}{cc} \int e^{2ik(|R_{s,z_{t}}|-|R_{s,z}|)}Q\left(w,s,z\right)A\left(w,s,z\right)dwds. & \propto \int e^{i[z_{t}-x_{0}\left(s\right)]\xi }d\xi \end{array}$$

Through a mathematical derivation, we can get
$$Q=\frac{\chi \left(w,s,z_{t}\right)\left|\frac{\partial (\xi )}{\partial (s,w)}\right|}{A(w,s,z_{t})},$$
which is called a filter or a weight function corresponding to the antenna in the frequency domain. Here, $\chi (w,s,z_{t})$ is a smooth cutoff function that prevents division by zero, and $\xi =2k\rho (z_{t}-x_{0}\left(s\right))$, with the operator ρ projecting a vector to get its first two components. Furthermore, we have
(6)$$I\left(z_{t}\right)=\;\int \int \;e^{-i[z_{t}-x_{0}\left(s\right)]\xi }d\xi V\left(z\right)dz$$
where $x_{0}\left(s\right)$ denotes the antenna location, and $z_{t}$ is the z-coordinate of the target. In the traditional BPA, the first step is range matching filtering. Then, the time delay of the doubled distance between the target and SAR platform is calculated to compensate for the phase factor. The last step is a summation of all signals to form the SAR image.

After the SAR image is formed, the image needs to be processed to provide for better image use. The image edge reflects the border information, which mainly refers to the discontinuous property in a certain region, such as a change in the gray value or a change in the texture structure. The image edge is still of great help to better utilize the image.

### 2.2. SAR Image Edge Detection

In SAR image edge detection, the first-order differential and second-order differential are usually computed to detect the edge. There are many popular second-order differential operators, such as those of Canny, Roberts, and so on, and they are widely used for image edge detection. However, because of the intensity inhomogeneities and the speckle noise of a SAR image, direct implementation of these algorithms usually results in too many false edges; therefore, they cannot be used directly on SAR images in practice. Here, we mainly analyze the gradient method, and we mostly use the first-order gradient, the second-order gradient Gauss–Laplace operator, and the Canny operator detection methods. The first-order gradient operator denotes vertical and horizontal gradients of the image. However, in practice, the edge direction is not always horizontal or vertical, so, the gradient of four directions is generally used, and the results are combined to get the final result. The second-order gradient operators combine Gaussian smoothing with the Laplacian algorithm, which is called the LoG operator. The LoG operator is obtained by combining the Gaussian-smoothed pretreatment result with the detection result. The Canny operator uses multiple steps to complete image edge extraction, and it is more complex than the other operators. Before using the Canny operator, it is necessary to perform Gaussian filtering to smooth the image to reduce the influence of noise on gradient extraction; then, the image gradient can be obtained. However, since a large number of false edges and repetitive edges are generated during the above calculation, it is very difficult to conduct subsequent image segmentation. Figure 1 and Figure 2 show the simulation results.

In order to detect the edge of a SAR image effectively, it is reasonable to enhance the edge by the first-order differential operator first and then seek for the proper algorithm to detect the edge. Moreover, here, we combined the first-order differential operator with the imaging operator to overcome the influence of speckle noise. In this way, we use the first-order filter more effectively and utilize the obtained SAR image more easily. SAR image edge detection is performed by employing the approach called the primal sketch [22].

Furthermore, to more effectively utilize the SAR data, many SAR applications perform SAR image segmentation first. Like SAR image edge detection, SAR image segmentation also suffers from many problems. The classic methods may fail with a SAR image due to its characteristics. Enhancing the edges of a SAR image using the BPA can contribute to SAR image segmentation. In this study, the method called the Chan-Vese (CV) level was employed to accomplish SAR image segmentation.

### 2.3. Edge Detection Using Multiscale Edge Features

SAR image edge detection is performed via a series of multiscale Gaussian filters through which we can get the edges for a specific scale:(7)Ł=f⊗g(:,:,t)
where *f* is the Gaussian filter; $t=\sigma^{2}$ is the bandwidth of the Gaussian filters, where σ is the standard deviation. Then, the gradient method is applied by combining the multiscale images. For a specific *t*, we seek the corresponding edges. Let $G_{-norm}L=t^{\gamma }\left(L_{x}^{2}+L_{y}^{2}\right)$ and γ=0.5. The specific value of γ used is derived from analytical edge models and the constraint that $G_{-norm}L$ is maximized at the characteristic scale of the edge. Then, the edge detection condition can be defined by
(8)$$\left\{\begin{array}{c} L_{vv}=0\\ L_{vvv}<0\\ \frac{\partial G_{-norm}L}{\partial t}=0\\ \frac{\partial G_{-norm}L}{\partial t^{2}}<0 \end{array}\right.$$
where *v* represents the gradient direction. For each edge in a single image, we can get the corresponding scale *t*, so the edge can be detected.

The detected multiscale edge features are then used to obtain the candidate lines, which are further refined to get the final detection result.

### 2.4. Set Level Method with Intensity Inhomogeneities

The set level methods for image segmentation have many successful applications. In order to deal with the SAR image, we added some SAR characteristics to it. The original energy function of the set level method can be described by the following [13]:(9)$$\begin{array}{cc} E^{CV}\left(c_{1},c_{2},C\right)= & \mu Length(C)\\ & +\lambda_{1}\int_{inside(C)}{\left|\mu_{0}\left(x,y\right)-c_{1}\right|}^{2}dxdy\\ & +\lambda_{2}\int_{outside(C)}{\left|\mu_{0}\left(x,y\right)-c_{2}\right|}^{2}dxdy \end{array}$$
where μ, $\lambda_{1}$, and $\lambda_{2}$ are all positive values used to balance different items; $c_{1}$ is the mean value of the foreground area; $c_{2}$ is the mean value of the background area; and Length is the length of the curve. The physics-related meaning of this equation is that a curve is used to separate the image into foreground and background areas, and the selected curve is the one that minimizes the function.

In order to optimize the energy function, the function can be rewritten as
(10)$$\begin{array}{cc} E^{CV}\left(c_{1},c_{2},C\right)= & \mu \int_{\Omega }\delta_{\varepsilon }\left(\phi \left(x,y\right)\right)\left|\nabla \phi (x,y)\right|dxdy\\ & +\lambda_{1}\int_{\Omega }{\left|\mu_{0}\left(x,y\right)-c_{1}\right|}^{2}H_{\varepsilon }\left(\phi \left(x,y\right)\right)dxdy\\ & +\lambda_{2}\int_{\Omega }{\left|\mu_{0}\left(x,y\right)-c_{2}\right|}^{2}\left(1-H_{\varepsilon }\left(\phi \left(x,y\right)\right)\right)dxdy \end{array}$$
where ϕ represents the zero-level set of a Lipschitz function, ∇ is a gradient operator, and H(z) and δ(z) are defined as
(11)$$H_{\varepsilon }\left(z\right)=\frac{1}{\pi }\arctan\left(\frac{z}{\varepsilon }\right)$$
(12)$$\delta_{\varepsilon }\left(z\right)=\frac{1}{\pi }\;\frac{\varepsilon }{\varepsilon^{2}+z^{2}}$$

Applying a gradient to Equation (Equation 10), we get
(13)$$\frac{\partial \phi }{\partial t}=\delta_{\varepsilon }\left(\phi \right)\left[\mu div\left(\frac{\nabla \phi }{\left|\nabla \phi \right|}\right)-\lambda_{1}{\left(\mu_{0}-c_{1}\right)}^{2}+\lambda_{2}{\left(\mu_{0}-c_{2}\right)}^{2}\right]$$
where div represents divergence. The final update functions are given by
(14)$$\begin{array}{c} c_{1}\left(\phi \right)=\frac{\int I\left(x,y\right)H_{\varepsilon }\left(\phi \left(x,y\right)\right)dxdy}{\int H_{\varepsilon }\left(\phi \left(x,y\right)\right)dxdy}\\ \;c_{2}\left(\phi \right)=\frac{\int I\left(x,y\right)[1-H_{\varepsilon }\left(\phi \left(x,y\right)\right)]dxdy}{\int [1-H_{\varepsilon }\left(\phi \left(x,y\right)\right)]dxdy} \end{array}$$

To deal with intensity inhomogeneities, a bias field is added. For a SAR image, the bias field is defined as follows:(15)I=bJ
where *J* is the true image without intensity inhomogeneities, and *b* is the bias field. We use the logarithmic transformation to change multiplication into addition. Considering the bias field model, Equation (Equation 10) can be rewritten as
(16)$$\begin{array}{cc} E^{CV}\left(c_{1},c_{2},C\right)= & \mu Length(C)\\ & +\lambda_{1}\int_{inside(C)}{\left|\mu_{0}\left(x,y\right)-b\left(x,y\right)c_{1}\right|}^{2}dxdy\\ & +\lambda_{2}\int_{outside(C)}{\left|\mu_{0}\left(x,y\right)-b\left(x,y\right)c_{2}\right|}^{2}dxdy \end{array}$$

Thus, the final update functions are defined by
(17)$$\begin{array}{c} c_{1}\left(\phi \right)=\frac{\int \left({\mu_{0}}^{{}^{\prime }}\left(x,y\right)-b\left(x,y\right)\right)H_{\varepsilon }\left(\phi \left(x,y\right)\right)dxdy}{\int H_{\varepsilon }\left(\phi \left(x,y\right)\right)dxdy}\\ \;c_{2}\left(\phi \right)=\frac{\int \left({\mu_{0}}^{{}^{\prime }}\left(x,y\right)-b\left(x,y\right)\right)\left[1-H_{\varepsilon }\left(\phi \left(x,y\right)\right)\right]dxdy}{\int [1-H_{\varepsilon }\left(\phi \left(x,y\right)\right)]dxdy}\\ b\left(x,y\right)=\int \left({\mu_{0}}^{{}^{\prime }}\left(x,y\right)-c_{1}\right)H_{\varepsilon }\left(\phi \left(x,y\right)\right)+\left({\mu_{0}}^{{}^{\prime }}\left(x,y\right)-c_{2}\right)\left(1-H_{\varepsilon }\left(\phi \left(x,y\right)\right)\right))dx \end{array}$$

## 3. New Method Combining Matched Filtering and Edge Enhancement

The differential operation should be performed on $I(z_{t})$ to get the image edge. By computing the differential of both sides of Equation (Equation 6), and through the evolution of partial differential equations, we get $Q^{*}$, which embodies the influence of the differential operation. By computing the differential of both sides of Equation (Equation 6), we get
(18)$$\nabla_{z}I\left(z_{t}\right)=\;\int \int \;\left(i\xi \right)*e^{-i(z_{t}-x_{0}\left(s\right))\xi }d\xi V\left(z\right)dz$$
where $\nabla_{z}$ represents the differential operator. Considering the direction of the SAR image, we can add the direction vector $\hat{u}$ to Equation (Equation 18)
(19)$$\hat{u}*\nabla_{z}I\left(z_{t}\right)=\;\int \int \;\hat{u}*\left(i\xi \right)*e^{-i(z_{t}-x_{0}\left(s\right))\xi }d\xi V\left(z\right)dz$$

Thus, we get the expression for $Q^{*}$:(20)$$Q^{*}=\hat{u}\xi *Q$$
where $\hat{u}$ is defined as
(21)$$\begin{array}{c} \hat{u}=\left[\begin{array}{c} {\hat{u}}_{p1}\\ {\hat{u}}_{p2} \end{array}\right]\\ {\hat{u}}_{p1}=\left[1,0\right]\\ {\hat{u}}_{p2}=\left[0,1\right] \end{array}$$

Therefore, the final SAR image expression is defined by
(22)$$\overset{⌢}{\mu }\cdot \nabla_{z}I\left(z_{t}\right)=\;\int \int \;\left[{\hat{u}}_{p1}+{\hat{u}}_{p2}\right]\cdot \left(i\xi \right)\cdot e^{-i(z-z_{t})\xi }d\xi V\left(z\right)dz$$

If the pixels of a target area are all the same, the edge can be obtained directly. However, the area has a gray change. So, by exploiting $Q^{*}$, we can enhance the edge of the SAR image by the imaging algorithm, which will contribute to SAR edge detection. By changing the direction vector $\hat{u}$, we may get either the azimuth direction or the range direction. Although the signal during imaging processing is complex-valued, a new operator can be added to the BPA directly. Therefore, Equation (Equation 22) is in the complex-valued form. In order to enhance the edge in all directions, we compute the differential operation in different directions, and, eventually, we synthesize the results into the final result. The image synthesis procedure uses both the range-direction result and the azimuth-direction result by adding them pixel by pixel.
(23)$$I=I_{range}+I_{azimuth}$$
where *I* is the final result, $I_{range}$ is the edge-enhanced result in the range direction, and $I_{azimuth}$ is the edge-enhanced result in the azimuth direction.

The detailed flowchart of the proposed algorithm is presented in Figure 3, and the corresponding procedures are presented in Table 1.

In practice, the algorithm is performed in two steps. Firstly, after range compression, the image is filtered by matched filtering in the range direction; then, the phase factor is compensated for according to the time delay. Secondly, the echo data is filtered by matched filtering in the azimuth direction, and during the compensation of the phase factor, the first-order filter is added in the azimuth direction. Finally, these two results are combined to form the final edge-enhanced SAR image.

Through the proposed imaging algorithm, which combines the gradient operation and two-dimensional focus, we can enhance the edges of SAR images. However, the edge-enhanced images still need further processing to get the edge. Therefore, here, a specific algorithm was chosen to accomplish edge detection and image segmentation.

## 4. Simulations and Analysis

In this section, the experiments performed using the proposed algorithm are presented, and the obtained experimental results are compared with the results obtained by the traditional imaging procedure. In order to validate the effectiveness of the proposed algorithm, the proposed algorithm was used on both a geometric image and a real scene. SAR image edge detection was carried out by utilizing the approach called the primal sketch. Also, SAR image segmentation was applied to the obtained edge-enhanced image via the CV level set method.

### 4.1. Edge-Enhanced Back-Projection Algorithm

The results of a rectangular area obtained by performing the procedure for only one direction—the range direction or the azimuth direction—are presented in Figure 4, wherein it can be seen that the edge in the respective direction is enhanced. The final imaging result of the proposed algorithm was synthesized from the two one-direction results, so the edges in both directions were enhanced.

To evaluate the results further, we performed edge detection and image segmentation.

### 4.2. SAR Edge Detection

Figure 5 demonstrates the imaging and edge detection results of the rectangular area. The edge of the rectangle formed by the new algorithm is more distinct. As for edge detection, both results can be accepted because the edge of the rectangular area can be detected easily. Figure 6, in the circular area, shows similar phenomena. Figure 7 is divided into nine blocks, each having different grayscales and graphics. Comparing Figure 7a,c, we can see that the edge of the SAR image generated by the proposed method is clearer. After edge detection, this advantage is more intuitive, especially for the edges inside the image. The red circle in Figure 7d marks areas that are absent from Figure 7b. This shows that the edge of the image generated by this algorithm is more easily detected. It is also reflected by the quantitative evaluation index in Table 2. The mean square error (MSE) has improved by 19.13% and Pratt’s figure of merit (FOM) has improved by 11.91%. However, at the same time, there is still the problem of incomplete edge detection in Figure 7, but this is also the difficulty for SAR image edge detection. The above-mentioned scenes were simulated for the sake of supervised evaluation. In order to validate the effectiveness of the proposed algorithm, we performed the algorithm for complicated real scenes. Figure 8 demonstrates the imaging and edge detection result of a complicated real area. The result of this scene has enhanced edges, which verifies our method as described. We can see that the edge detection result shows that the edge-enhanced imaging result does contribute to edge detection. Even in a complicated real scene, the edge is well detected compared to the edge detection result of the traditional method. For example, the edge in the red box in Figure 8d is more complete than that in Figure 8b. Figure 9 shows similar results.

To evaluate the edge detection of the simulated geometry, we used the concept of the mean square error and quality factor. The mean square error measures the difference between two images and is defined as
(24)$$MSE=\frac{1}{MN}{\left[X(i,j)-Y(i,j)\right]}^{2}$$
where X(i,j) is the real edge of the image, Y(i,j) is the detection edge, and *M* and *N* are the column and row of the image. The quality factor is defined as
(25)$$FOM=\frac{1}{\max(X_{N},Y_{N})}\sum_{i=1}^{Y_{N}}\frac{1}{1+\alpha d_{i}^{2}}$$
where $X_{N}$ is the number of edge points in the reference image, $Y_{N}$ is the number of edge points in the detection result, α is a constant, and *d* is the distance between the detected edge point and the true edge point. The physical meaning of *d* is the geometric distance between the reference point and the actual test point.

Table 2 shows the evaluation of the edge detection process using the supervised ground truth (the true edge of the image). It can be seen that the edge detection results obtained by the algorithm proposed in this section are significantly superior to the traditional BPA. For the rectangular area, the edge is easily detected and the edge-enhanced images achieve the same result compared to the ordinary one. For the circular area, the edge is a little bit difficult, and the edge-enhanced images have better results.

For the simulated scenes, we know exactly where the true edge is, so we used the ground truth to evaluate the edge detection results. However, for real echo data, we do not have the ground truth. In order to evaluate the edge detection result, we used the concept of continuity and reconstruction similarity to denote the quality of the edge. Integrity is defined to describe the continuity of the specific detected edge, and continuity is defined as
(26)$$SC^{i}=S\left(C^{i}\right)=2\times \left[\frac{1}{1+\exp(-C^{i}/\alpha )}-0.5\right]$$
where
(27)$$C^{i}=\sum_{k=1}^{n_{i}}c_{k}^{i}\;$$
(28)$$c_{k}^{i}=\left\{\begin{array}{c} \frac{d_{k}^{i}}{D}\;\;d_{k}^{i}<D\;\\ \;1\;\;d_{k}^{i}<D \end{array}\right.$$
and $d_{k}^{i}$ is the distance between the pixels at the edge and the center.

Reconstruction similarity is used to compare the original image and the image reconstructed from the edge image. An interpolation algorithm was first implemented to reconstruct the image using the edge detection result. The similarity is defined to compare the differences between two images using structural similarity (SSIM). Similarity is defined as
(29)$$s\left(X,Y\right)=\frac{\sigma_{XY}+C_{3}}{\sigma_{X}+\sigma_{Y}+C_{3}}$$
where $\sigma_{X}$ and $\sigma_{Y}$ represent the covariance of each image, respectively, and $\sigma_{XY}$ represents the cross-covariance of the two images. $C_{3}$ is a constant.

Table 3 is the result of the unsupervised evaluation for the real SAR images. From the data, we can see that the edge detection result of the edge-enhanced images shows a better performance for continuity and integrity. For the complicated scene, edge detection is much more difficult, and the edge detection of the edge-enhanced image is much better than that of ordinary method’s result.

### 4.3. SAR Image Segmentation

In order to further evaluate the effectiveness of the proposed imaging algorithm, we performed SAR image segmentation of the obtained SAR images to assess the improvement of the segmentation result. Similar to above, we used both simulated scenes and real scenes.

In Figure 10 and Figure 11, we can see that both SAR images are well segmented, but we can see in (d) that the edge is better than that in (c). In Figure 12, it is clear that both the edge and the result in (d) are better than that in (c). Because we have the ground truth of the simulated area, we evaluated these segmentation results using the probabilistic Rand index (PRI), variability index (VI), and global consistency error (GCE). PRI compares the similarity edges between two images, so the bigger the PRI, the better the result. However, VI and GCE measure the difference between two images, so, the smaller the values, the better the results. Let *n* denote the number of pixels in an image. PRI is defined as
(30)$$\begin{array}{c} PRI\left(S,S_{test}\right)=\frac{1}{c_{n}^{2}}\sum_{i}\sum_{j\neq i}I(l_{i}=l_{j}\&\&l_{i}^{{}^{\prime }}=l_{j}^{{}^{\prime }})+I\left(l_{i}\neq l_{j}\&\&l_{i}^{{}^{\prime }}\neq l_{j}^{{}^{\prime }}\right) \end{array}$$
where && represents the logic and operator; $l_{i},l_{j}$ represent two pixels in the ground truth result; and $l_{i}^{{}^{\prime }},l_{j}^{{}^{\prime }}$ represent two pixels in the detected result. VI is defined as
(31)$$VI\left(S,S_{test}\right)=H\left(S\right)+H\left(S_{test}\right)-2I\left(S,S_{test}\right)$$
where *S* represents the true edge and $S_{test}$ represents the detected edge. GCE is defined as
(32)$$\begin{array}{c} GCE\left(S,S_{test}\right)=\frac{1}{n}\left[\min\left(\sum E(S^{K},S_{test}^{K^{{}^{\prime }}},p_{i}),\sum E(S_{test}^{K^{{}^{\prime }}},S^{K},p_{i})\right)\right] \end{array}$$
where $E(S^{K},S_{test}^{K^{{}^{\prime }}},p_{i})$ is defined as
(33)$$E\left(S^{K},S_{test}^{K^{{}^{\prime }}},p_{i}\right)=\frac{<R\left(S^{K},p_{i}\right)\setminus R\left(S_{test}^{K^{{}^{\prime }}},p_{i}\right)>}{<R(S^{K},p_{i})>}$$
and $<R(S^{K},p_{i})>$ are the elements in the set *R*; ∖ is the difference set of two sets; and $p_{i}$ is a pixel.

Table 4 is the evaluation result of SAR image segmentation. From Table 4, we can see that the proposed algorithm has improved the effect of SAR image segmentation. The improvement in PRI, VI, and GCE are more than 1.18%, 11.2%, and 11.72%, respectively.

## 5. Discussion

Focusing on the usage of SAR images, the new method described in this paper tends to improve SAR edge detection and image segmentation, which are fundamental procedures for SAR image processing. Instead of analyzing the SAR image processing algorithm, this method combines the gradient operation with the SAR imaging process, resulting in an edge-enhanced SAR image. All the operations proceed in the imaging algorithm. The advantages can be described as follows:The gradient operation is fused during the SAR image formation, which can overcome the problems related to speckle noise and intensity inhomogeneities to some extent, which is better than adding the operator directly to the SAR image;Because all the operations are used in the imaging procedure, we can use all the promising SAR edge detection and image segmentation methods to process the edge-enhanced SAR images, which may make many existing algorithms more powerful;We selected SAR edge detection and image segmentation methods to process the edge-enhanced SAR images, showing that the edge-enhanced SAR images can further improve these methods.

Despite the advantages mentioned above, there are some limitations to the proposed algorithm. Because the imaging procedure changes, the proposed algorithm may change the original SAR images, which may be unexpected. However, the proposed algorithm aims to better utilize SAR images, so we can use the edge detection result or image segmentation result as a mask to add to the traditional imaging result. In this way, this unexpected effect can be eliminated.

## 6. Conclusions

Characteristics of SAR present certain difficulties during the post-processing of a SAR image. In this paper, a new imaging algorithm is proposed to process SAR images during the imaging process, which can alleviate the difficulty of SAR image interpretation. The new imaging algorithm contributes to edge detection and image segmentation via the first-order gradient operation in the imaging procedure, which can overcome the problems of speckle noise and intensity inhomogeneities to some extent. As a consequence, the edge of the SAR images is enhanced after the imaging process. Both geometry graphics and real scenes were used for experiments to validate the new approach, which combines SAR imaging and SAR image processing. Edge detection and image segmentation were performed, and the results were evaluated to validate the effectiveness of the proposed algorithm. Experimental results show that processing the edge-enhanced SAR image with the edge detection and image segmentation algorithms can further improve the processing results. The edge detection evaluation indexes of our method have 19.13% and 11.91% improvement on MSE and FOM for the simulated complex graphics scene. And for the real scene, our method has improved more than 9.41% on the edge detection evaluation index for continuity and 3.8% for reconstruction similarity. Finally, for the simulated scene, the supervised image segmentation indexes of our method have more than 1.18%, 11.2% and 11.72% improvement on PRI, VI, and GCE. Furthermore, the idea of incorporating imaging processing algorithms into imaging algorithms can be instructive for improvements in SAR imaging algorithms.

## Figures and Tables

**Figure 1 sensors-18-04133-f001:**
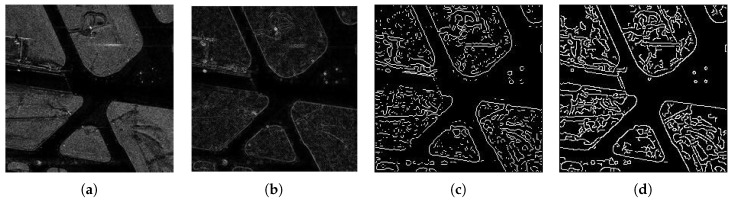
The edge detection results of synthetic aperture radar (SAR) image1. (**a**) Original image; (**b**) first-order result; (**c**) Gaussian–Laplacian operator result; (**d**) Canny operator result.

**Figure 2 sensors-18-04133-f002:**
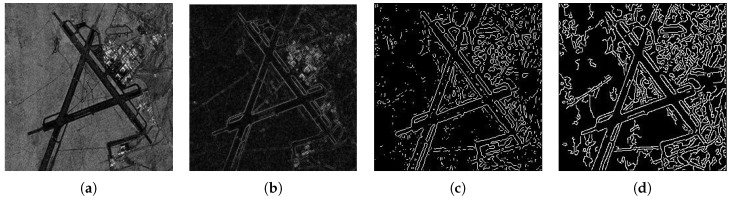
The edge detection results of SAR image2. (**a**) Original image; (**b**) first-order result; (**c**) Gaussian–Laplacian operator result; (**d**) Canny operator result.

**Figure 3 sensors-18-04133-f003:**
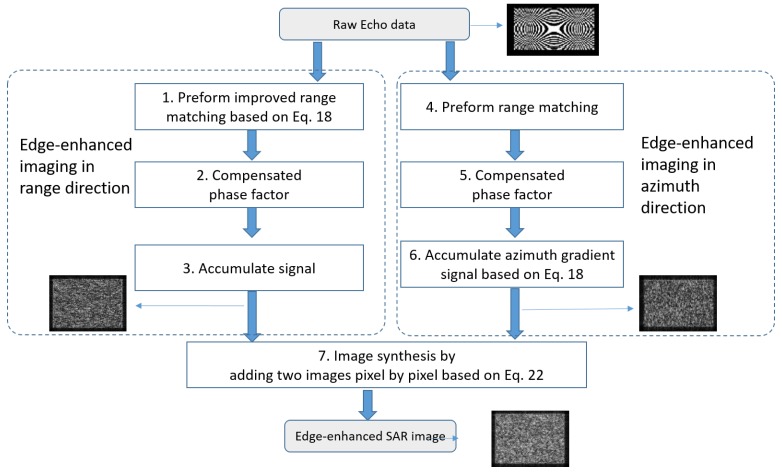
The flowchart of the proposed algorithm.

**Figure 4 sensors-18-04133-f004:**
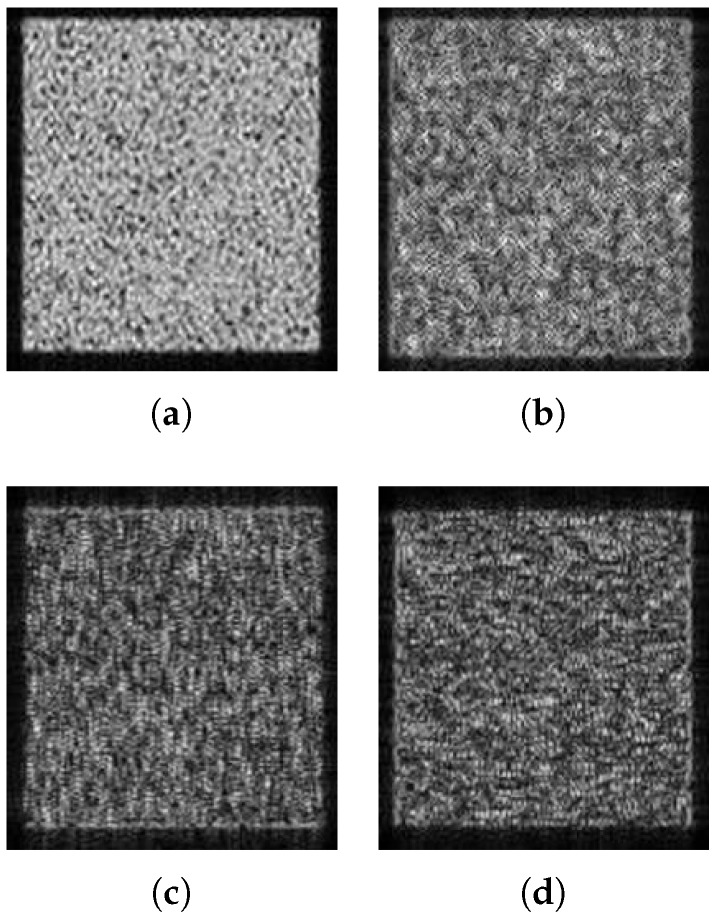
The imaging results of the rectangular areas. (**a**) Original image; (**b**) edge-enhanced SAR image; (**c**) azimuth direction edge-enhanced SAR image; (**d**) range direction edge-enhanced SAR image.

**Figure 5 sensors-18-04133-f005:**
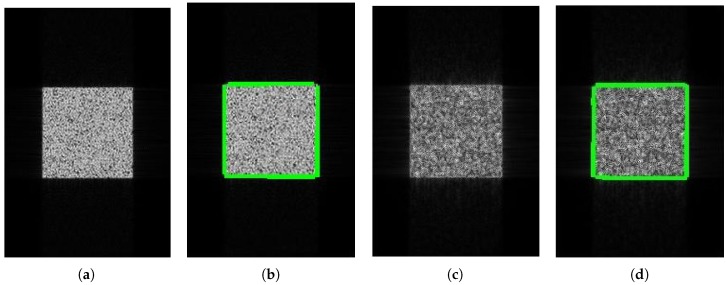
The imaging and edge detection result for the rectangular area. (**a**) Original image; (**b**) edge detection result of the original image; (**c**) edge-enhanced SAR image; (**d**) edge detection result of the edge-enhanced SAR image.

**Figure 6 sensors-18-04133-f006:**
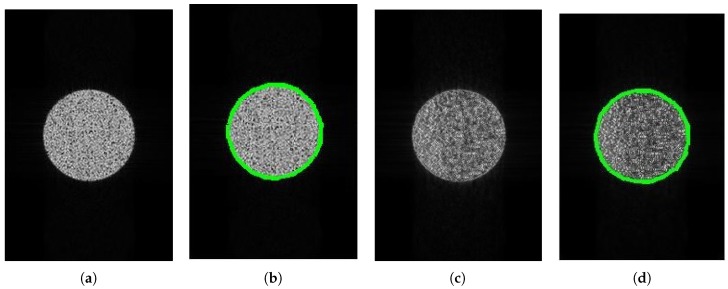
The imaging and edge detection result for the circular area. (**a**) Original image; (**b**) edge detection result of the original image; (**c**) edge-enhanced SAR image; (**d**) edge detection result of the edge-enhanced SAR image.

**Figure 7 sensors-18-04133-f007:**
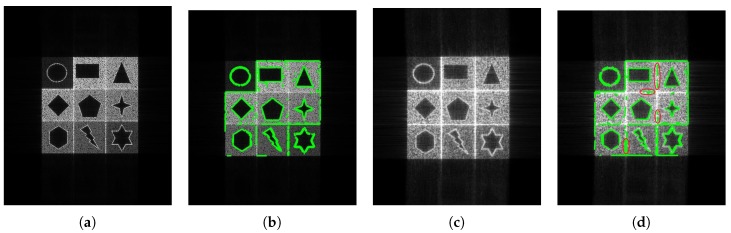
The imaging and edge detection result of complex graphics area. (**a**) Original image; (**b**) edge detection result of the original image; (**c**) edge-enhanced SAR image; (**d**) edge detection result of the edge-enhanced SAR image.

**Figure 8 sensors-18-04133-f008:**
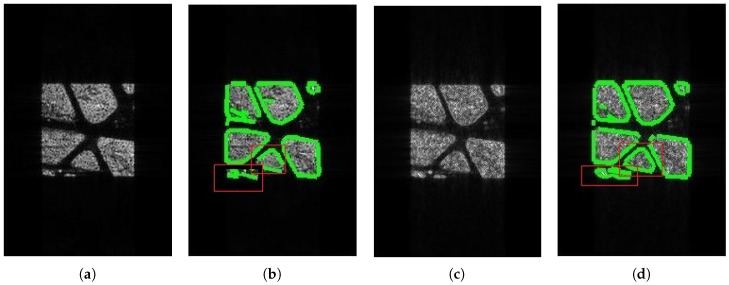
The imaging and edge detection result of complex area1. (**a**) Original image; (**b**) edge detection result of the original image; (**c**) edge-enhanced SAR image; (**d**) edge detection result of the edge-enhanced SAR image.

**Figure 9 sensors-18-04133-f009:**
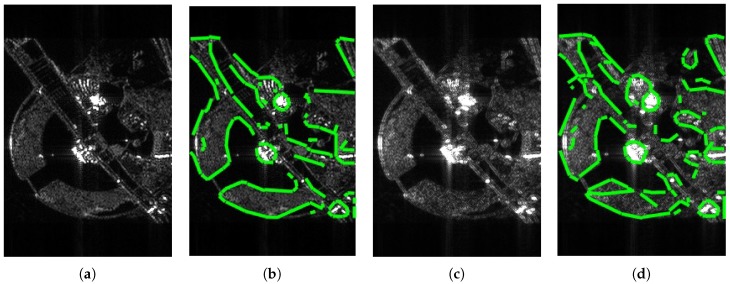
The imaging and edge detection result of complex area2. (**a**) Original image; (**b**) edge detection result of the original image; (**c**) edge-enhanced SAR image; (**d**) edge detection result of the edge-enhanced SAR image.

**Figure 10 sensors-18-04133-f010:**
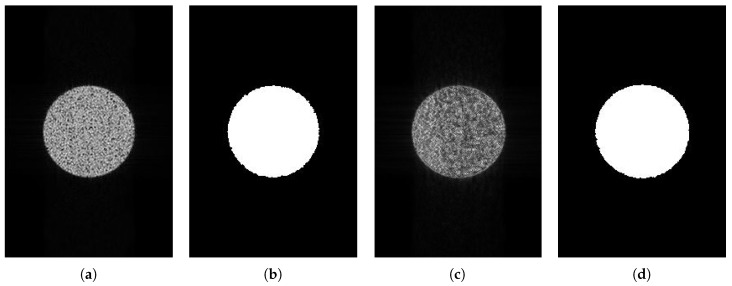
The imaging and image segmentation result of the circular area. (**a**) Original image; (**b**) image segmentation result of the original image; (**c**) edge-enhanced SAR image; (**d**) image segmentation result of the edge-enhanced SAR image.

**Figure 11 sensors-18-04133-f011:**
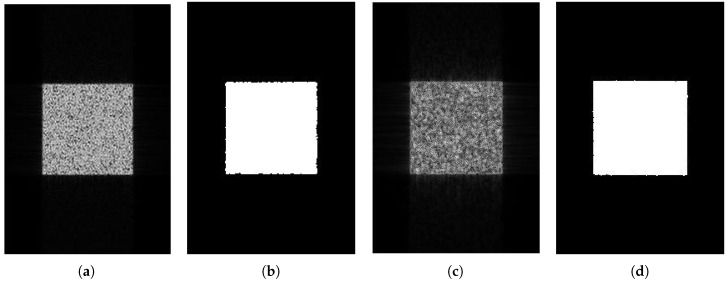
The imaging and image segmentation result of the rectangular area. (**a**) Original image; (**b**) image segmentation result of the original image; (**c**) edge-enhanced SAR image; (**d**) image segmentation result of the edge-enhanced SAR image.

**Figure 12 sensors-18-04133-f012:**
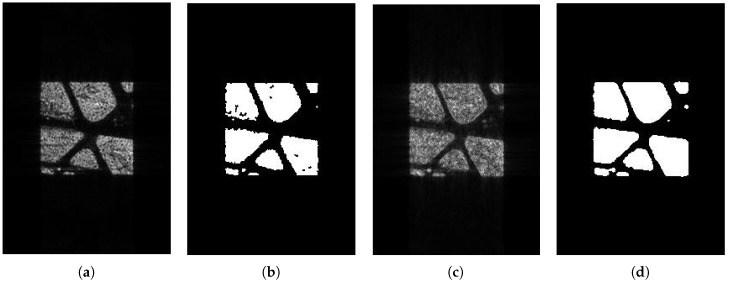
The imaging and image segmentation result of the complicated area. (**a**) Original image; (**b**) image segmentation result of the original image; (**c**) edge-enhanced SAR image; (**d**) image segmentation result of the edge-enhanced SAR image.

**Table 1 sensors-18-04133-t001:** Algorithm procedures.

Procedure	Method
1	Input the raw echo data and perform the improved range matching based on Equation (Equation 18) in the range direction using $\hat{u}={\hat{u}}_{p1}$.
2	Compensate for the phase factor in the range direction.
3	Accumulate the signal in the range direction, and output the edge-enhanced image in the range direction $I_{range}$.
4	Input the raw echo data and perform range matching in the azimuth direction.
5	Compensate for the phase factor in the azimuth direction.
6	Accumulate the azimuth gradient signal based on Equation (Equation 18) in the azimuth direction using $\hat{u}={\hat{u}}_{p2}$ and output the edge-enhanced image in the range direction $I_{azimuth}$.
7	Synthesize the image by adding two images pixel by pixel based on Equation (Equation 22), then output the edge-enhanced SAR image *I*.

**Table 2 sensors-18-04133-t002:** Supervised evaluation of edge detection. Mean square error (MSE), Pratt’s figure of merit (FOM).

Image	MSE	Improvement	FOM	Improvement
Rectangular area	0.0027	/	0.98	/
Edge-enhanced rectangle	0.0027	0.00%	0.98	0.00%
Circular area	0.0353	/	0.55	/
Edge-enhanced circular	0.0042	88.10%	0.98	78.18%
Complex graphics	0.0115	/	0.42	/
Edge-enhanced complex graphics	0.0093	19.13%	0.47	11.91%

**Table 3 sensors-18-04133-t003:** Unsupervised evaluation of edge detection.

Image	Continuity	Improvement	Reconstruction Similarity	Improvement
Complicated scene1	0.85	/	0.79	/
Edge-enhanced complicated scene1	0.93	9.41%	0.82	3.8%
Complicated scene2	0.80	/	0.78	/
Edge-enhanced complicated scene2	0.89	11.25%	0.81	12.5%

**Table 4 sensors-18-04133-t004:** Supervised evaluation of image segmentation. Probabilistic Rand index (PRI), variability index (VI), and global consistency error (GCE).

Image	PRI	Improvement	VI	Improvement	GCE	Improvement
Circular scene	0.9477	/	0.5795	/	0.0534	/
Edge-enhanced circular scene	0.9589	1.18%	0.5215	11.2%	0.0478	11.72%
Rectangular scene	0.9595	/	0.5348	/	0.0546	/
Edge-enhanced rectangular scene	0.9680	8.5%	0.4598	16.31%	0.0449	17.77%

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
