# Peer review of "A New Synthetic Aperture Radar (SAR) Imaging Method Combining Match Filter Imaging and Image Edge Enhancement"

_sensors, 2018, doi:10.3390/s18124133_

Reviewer 1 Report

The paper is now interesting. Some simulations, more reasonable, have been considered. I only ask the authors to write (better) the conclusion section trying to make it more evident the reason why some readers should use this method instead of other equivalent approaches.

Author Response

Response to Reviewer 1 Comments

Point 1: The paper is now interesting. Some simulations, more reasonable, have been considered. I only ask the authors to write (better) the conclusion section trying to make it more evident the reason why some readers should use this method instead of other equivalent approaches.

Response 1: Thank you so much for your advice. We have modified our conclusion in the reversion. In the reversion, we have emphasized the difficulty of SAR image interpretation and explained the proposed algorithm can relieve the difficulty of SAR image interpretation especially for the problems of speckle noise and the intensity inhomogeneities. Additionally, we have described that experimental results show that process the edge-enhanced SAR image with edge detection and image segmentation algorithms can further improves the process results. Finally, we have emphasized the new algorithm this paper proposed may make many existing algorithms for post processing of SAR image more powerful and the idea of incorporating imaging processing algorithms into imaging algorithms can be instructive for improvements in many existing SAR imaging algorithms.

Point 2: Moderate English changes required.

Response 2: We have used a professional English editing service to improve the English writing, also the authors have checked and corrected all the equations in this version. We expect this revision can convince the reviewers and meet the requirements of this journal. The certificates of English Editing are shown in the following pdf. The first is an English certificate of the submitted version, which was issued on March 08, 2018 and the second is an English certificate of  reversion, which was issued on November 01, 2018.

Reviewer 2 Report

An original approach that deserves the attention of the Journal audience.

Author Response

Response to Reviewer 2 Comments

Point 1: In the paper the SAR scenario under consideration has to be presented graphically. All geometry parameters have to be defined. Just after formula (2) all parameters have to be defined: Zt , the point target position in fast time domain,ω -the frequency from the spectrum, y – as a coordinate.

Explain I(Zt) in formula (3) and its relation to V(z). Define the relationship between d(t,s) and D(s,ω).

In formula (3) explain parameter γ. Cite the source of formula (3). In formula (3) V(z) has to be before dω.ds.dz. Explain dz after substitution D(s,ω) with A(ω,s,z)V(z).

Row 101: 2.3. Start with a capital letter and everywhere in the text where it is necessary.

Explain bandwidth t = σ2, where σ2 is the dispersion, and γ?

Row 106: 2.4. Start with a capital letter.

Cite the source of the formula (8). In (9) define φ and nabla (φ).

In (12) define μdiv.

In table 1: row 2 write “Compensate the phase factor in the range direction”, row 5“Compensate the phase factor in the azimuth direction”. (In point of view of the reviewer).

Page 9: Figure 5, start with a capital letter and everywhere it is necessary.

Row 197: what is “dege”. Most probably it a “edge”.

The English language in the paper has to be improved.

Response 1: Thank you so much for your kind comment and advice.

We have modified the English grammar error in the article and added the meaning of some variables in the formula according to your review.

1. Every chapter and section have started with a capital letter.

2. Zt, ω and y of formula (2) have been defined. I(Zt) and the relationship between d(t,s) and D(s,ω) have been explained in formula (2).

3. We have added a new formula (formula (3)) to describe the relationship between I(Zt) and V(z).

4. We have cited the source of formula (4) (original formula (3)) and changed the location of V(z). From the formula (3), we can know the meaning of dz has not changed.

5. In formula (7), σ and γ have been defined. σ is the standard deviation and the specific value of γ used is derived from analytical edge models and the constraint that G-normL is maximized at the characteristic scale of the edge.

6. We have cited the source of formula (9) (original formula (8)).

7. We have defined φ and nabla (φ) in formula (10) (original formula (9)) and div in formula (11) (original formula (10)) has also been defined. Otherwise, μ in formula (11) has already been defined in formula (9).

8. We have modified the row 2 and row 5 in Table 1. (From “Compensated” to “Compensate”) and corrected spelling error in Row 197.

Point 2: English language and style are fine/minor spell check required.

Response 2: We have used a professional English editing service to improve the English writing, also the authors have checked and corrected all the equations in this version. We expect this revision can convince the reviewers and meet the requirements of this journal.The certificates of English Editing are shown in the following pdf. The first is an English certificate of the submitted version, which was issued on March 08, 2018 and the second is an English certificate of reversion, which was issued on November 01, 2018.
